# Combined Soluble Fiber-Mediated Intestinal Microbiota Improve Insulin Sensitivity of Obese Mice

**DOI:** 10.3390/nu12020351

**Published:** 2020-01-29

**Authors:** Chuanhui Xu, Jianhua Liu, Jianwei Gao, Xiaoyu Wu, Chenbin Cui, Hongkui Wei, Rong Zheng, Jian Peng

**Affiliations:** 1Department of Animal Nutrition and Feed Science, College of Animal Science and Technology, Huazhong Agricultural University, Wuhan 430070, China; xuchuanhui001@webmail.hzau.edu.cn (C.X.); 13586414898@163.com (X.W.); cuichenbin@webmail.hzau.edu.cn (C.C.); weihongkui@mail.hzau.edu.cn (H.W.); pengjian@mail.hzau.edu.cn (J.P.); 2Department of Animal Genetics and Breeding, College of Animal Science and Technology, Huazhong Agricultural University, Wuhan 430070, China; ljh1558985906@163.com (J.L.); 18335480573@163.com (J.G.); 3The Cooperative Innovation Centre for Sustainable Pig Production, Wuhan 430070, China

**Keywords:** obesity, insulin sensitivity, systemic inflammation, combined soluble fiber, intestinal microbiota

## Abstract

Dietary fiber, an important regulator of intestinal microbiota, is a promising tool for preventing obesity and related metabolic disorders. However, the functional links between dietary fiber, intestinal microbiota, and obesity phenotype are still not fully understood. Combined soluble fiber (CSF) is a synthetic mixture of polysaccharides and displays high viscosity, water-binding capacity, swelling capacity, and fermentability. We found that supplementing high-fat diet (HFD) with 6% CSF significantly improved the insulin sensitivity of obese mice without affecting their body weight. Replacing the HFD with normal chow basal diet (NCD), the presence of CSF in the feed significantly enhanced satiety, decreased energy intake, promoted weight and fat loss, and augmented insulin sensitivity. CSF also improved the intestinal morphological integrity, attenuated systemic inflammation, promoted intestinal microbiota homeostasis, and stabilized the production of short-chain fatty acids (SCFAs) that was perturbed during HFD-induced obesity, and these stabilizing effects were more prominent when the basal diet was switched to NCD. The enrichment of bacteria of the S24-7 family and *Allobaculum* genus increased markedly in the intestine following 6% CSF supplementation- and correlated with decreased adiposity and insulin resistance. Five bacterial genera that were decreased by CSF, including *Oscillospira*, unclassified Lachonospitaceae, unclassified Clostridiales, unclassified Desulfovibrionaceae, and unclassified Ruminococcae, were subjected to co-occurrence network analysis and were positively correlated to adiposity and insulin resistance, indicating a key role in the microbial response to CSF. Thus, CSF has a potential to promote insulin sensitivity and even reduce obesity via beneficial regulation of the gut microecosystem.

## 1. Introduction

Obesity is a global public health problem, with roughly 1.9 billion overweight and 600 million obese adults worldwide as of 2014 [1]. Obese individuals often suffer from severe metabolic disorders, which is a central player in the pathophysiology of diabetes mellitus, insulin resistance, dyslipidemia, hypertension, and atherosclerosis [2]. Studies have increasingly demonstrated that the gut microbiota play a central role in the development of obesity and metabolic disorders [3,4]. Individuals with excessive adipose deposition, insulin resistance, and dyslipidemia have low bacterial richness in their intestines [5]. The relative proportion of Bacteroidetes is often decreased in obese individuals and can be restored by low-calorie diet and weight loss [6]. An altered gut microbiome significantly affects host metabolism and physiological status through several pathways [7]. The intestinal bacteria interact with the intestinal epithelial cells and immune cells to strengthen the intestinal barrier function [8,9]. Intestinal dysbiosis induced by a high-fat diet (HFD) increases calorie retention [10] and intestinal permeability, promotes the release of lipopolysaccharides (LPS) into the bloodstream, and triggers an inflammatory response [11]. Therefore, restoring the intestinal microbiota is a viable strategy for the prevention and auxiliary treatment of obesity and related metabolic disorders.

Increased dietary fiber intake can effectively prevent HFD-induced obesity and metabolic disorders in both healthy [12,13,14,15] as well as obese individuals [16]. The degradation of dietary fiber in the gut is associated with an increase in the proportion of beneficial bacteria like *Bifidobacteria*, *Lactobacilli* and *Faecalibacterium prausnitzii*, which play an important role in regulating obesity and metabolic parameters [17,18]. For instance, *Bifidobacteria* reverses metabolic endotoxemia, and improves gut integrity and the associated metabolic changes in mice [19,20]. Dietary fiber is fermented by the gut microbes into short-chain fatty acids (SCFAs), which have beneficial effects on the gut barrier and mitigate obesity by regulating endocrine activity [21]. Butyrate, in particular, is a key promoter of colonic health and integrity, and meets 60%–70% of the energy requirements of colonocytes necessary for their proliferation and differentiation [22]. Furthermore, SCFAs stimulate the production of gut anorectic hormones such as glucagon-like peptide 1 (GLP-1) and peptide YY (PYY) from the L cells and decrease appetite [23]. However, there is evidence that SCFAs can also promote obesity by acting as the energy substrate [10], and obese individuals have higher SCFAs levels than those with a healthy BMI [24], which contradicts their supposed beneficial effects.

Konjac flour is a readily fermentable dietary fiber with high viscosity, water-binding capacity, and swelling capacity. The konjac-mannan polysaccharide protects against obesity [25] and metabolic disorders [26]. In our previous works, dietary supplementation with konjac flour improved insulin sensitivity and reproductive performance of pregnant sows [27]. However, the annual production of konjac flour is low and its price is high, which makes widespread use limited. Studies show that the mixture of hydrophilic colloid and pre-gelatinized starch can synergistically improve viscosity, water binding capacity, and stability [28,29]. In our previous works, combining guar gum, a strongly hydrophilic straight-chain β-1-4-mannose, with pre-gelatinized waxy maize starch resulted in a combined soluble fiber (CSF) with physicochemical properties similar to that of konjac flour [30,31]. In the present study, we determined the effects of this CSF on HFD-fed obese mice, and explored the relationship between dietary CSF consumption, intestinal microbiota, and metabolic status.

## 2. Materials and Methods

### 2.1. Animal Diets

The composition of all purified diets used in this study are listed in Appendix A. The energy content of the low-fat basal diet (LFD) was 3.6 kcal/g, with 19% kilocalories (kcals) from protein, 71% from carbohydrates, and 10% from fat. The calorific value of high-fat basal diet (HFD) was 5 kcal/g, with 19.4% of the calories from protein, 20.6% from carbohydrate, and 60% from fat. The normal chow diet (NCD) derived 14.1% of the calories from protein, 75.9% from carbohydrates, and 10% from fat, for a total energy content of 3.6 kcal/g. All feeds were purchased from Trophic Animal Feed High-Tech Corp. Ltd. (Jiangsu, China). The basal feed was supplemented with a suitable amount of CSF (14.3% guar gum (Shangdong Yunzhou Science and Technology Corp. Ltd., Yunzhou, China) and 85.7% pregelatinized waxy maize starch (Hangzhou Puluoxiang Starch Corp., Ltd., Hangzhou, China)) to replace cellulose.

### 2.2. Establishment of Obesity Model and Feeding Regimen

Specific pathogen free 4-to 5-week-old male C57BL/6 mice were purchased from Laboratory Animal Center, Huazhong Agricultural University, Wuhan, People’s Republic of China, and housed at 22-24 °C under a 12 h light/ dark diurnal cycle with food and water provided ad libitum. After a 7-day adaptation period, the mice were fed with LFD or HFD for 10 weeks. The mice were weighed, and the obese mice were identified as those with at least 20% weight gain compared to the LFD-fed mice. Twenty-one diet-induced obese (DIO) mice were randomly divided into the HF-C, 4% CSF and 6% CSF groups (n = 7/group), and fed HFD or CSF-supplemented HFD for 12 weeks (period 1). A control group of lean mice was included and fed LFD during the same period (LF-C, n = 7). In the second treatment period lasting 4 weeks, the animals were fed NCD or the corresponding adjusted diet. Food intake was measured daily and body weight was measured weekly. The animal experiments were approved by the Animal Care and Use Committee of the Huazhong Agricultural University (Wuhan, China) (HZAUMO-2018-047).

### 2.3. Glucose and Insulin Tolerance Tests

A glucose tolerance test (GTT) and an insulin tolerance test (ITT) were conducted at the end of the 12th and 16th weeks. For the GTT, the mice were fasted overnight, and the baseline blood glucose levels were measured using the Accu-Chek Active Blood Glucose Meter. The mice were then intraperitoneally injected with 2 mg glucose/g body weight, and blood glucose levels were measured 15, 30, 45, 60, and 120 min later. Two days later, the animals were fasted for 6 h, and the fasting glucose levels were determined as above. The ITT was initiated in these 6 h-fasted mice by injecting them with 1U insulin/kg body weight, and the blood glucose levels were measured 15, 30, 45, and 60 min after injection.

### 2.4. Serum Lipid Profile Analysis

Two days after the ITT, the mice were fasted overnight, and blood was collected from the tail vein under negative pressure conditions. After leaving the blood samples undisturbed for 2 h on ice, they were centrifuged at 1500× *g* for 10 min at 4 °C, and the sera were collected and stored at −80 °C until analysis. The levels of triglycerides (TG), non-esterified fatty acids (NEFA), low density lipoprotein cholesterol (LDL-C) and high-density lipoprotein cholesterol (HDL-C) were measured using commercial kits (Nanjing Jiancheng Bioengineering Institute, Nanjing, China). 

### 2.5. Histopathological Examination

After measuring the body weights at the end of the 16-week period, the mice were fasted for 6 h and then sacrificed. The liver and fat pads (inguinal, interscapular, epididymal, and peri-renal white adipose tissues, and interscapular brown adipose tissue) were collected and weighed. The liver pieces were fixed in 4% paraformaldehyde for 3 days, embedded in paraffin, cut into 4 μm-thick sections, and stained with oil red O. The epididymal adipose tissue was also processed as above, and its sections were stained with hematoxylin and eosin (H&E). The fat content in the liver and adipose cells size were determined using Image Pro Plus software. Colonic tissues were harvested and stained with H&E as above. The crypt length and mucosal layer thickness were determined using Image Pro Plus.

### 2.6. Analysis of Feeding Behavior and Satiety

Food intake was recorded daily, and the gross energy content of the diets was measured using the IKA C 3000 h eco Package calorimeter. The accumulative energy intake was then calculated by multiplying the total food intake with the gross energy content. During the 6th and 15th week (i.e., mid-point of each feeding period), the 24 h feeding pattern was evaluated by recording food consumption between 0–1, 1–2, 2–4, and 4–24 h after overnight fasting. One day later, the 2 h postprandial blood was collected, and the levels of the satiety hormones PYY and GLP-1 in serum samples were measured using commercial kits (Jiyinmei Biotechnology Co. Ltd., Wuhan, China).

### 2.7. Biochemical Tests

Serum samples were collected from the euthanized mice at the end of 16 weeks. Chromogenic assay kits were used to analyze the levels of lipopolysaccharide (LPS), leptin, adiponectin (Jiyinmei Biotechnology Co., Ltd., Wuhan, China), TNF-α, and IL-6 (Bioswamp Biotechnology Co., Ltd., Wuhan, China). The key resources of critical commercial and assay kits were provided in Appendix A.

### 2.8. In Vivo BrdU Staining

The differentially fed mice were given four intraperitoneal injections of 50 μg BrdU/g body weight with a 2 h interval between two consecutive injections. Twenty-four hours after the last injection, the mice were euthanized, and the proximal colon tissues were harvested, paraffin embedded, sectioned, and stained for BrdU. The BrdU-labeled nuclei were counted in at least three fields per slide.

### 2.9. RT-PCR

Total RNA was extracted from transverse colonic tissues using TRIzol (Invitrogen, Carlsbad, CA, USA) according to the manufacturer’s protocol. The expression levels of ZO-1, Occluding, and Claudin-2 mRNAs were analyzed by quantitative RT–PCR using specific primers (Appendix A). The expression levels of each were normalized to that of the housekeeping gene 36B4.

### 2.10. Gas Chromatography

The fecal levels of acetate, propionate, butyrate, isobutyrate, valerate, and isovalerate in the different mice were measured by gas chromatography. Briefly, approximately 10–30 mg feces from each mouse was homogenized in 500 μL methanol, ground to a fine powder using a grinding mill at 65 HZ for 120 s, and then vortexed for 30 s. The samples were centrifuged at 12,000 rpm and 4 °C for 15 min, and 400 μL supernatant from each sample was concentrated by centrifuging again. Each sample was re-dissolved in 50 μL methanol and centrifuged at 12,000 rpm and 4 °C for 15 min. One microliter of each sample was subjected to gas chromatography (GC 2010, Shimadzu, Japan equipped with a CP-Wax 52 CB column of 30.0 m × 0.53 mm i.d., Chrompack, Netherlands). SCFAs were quantified using standard curves of 0.5 to 100 µM organic acids (Fluka, Buchs, Switzerland).

### 2.11. Microbiome Sequencing

Feces samples were collected before, and after 12 and 16 weeks of the feeding regimen. Microbiome composition was determined by 16S rRNA gene amplification and sequencing as previously described (Chassaing et al., 2015). Briefly, DNA was extracted from feces and intestinal contents using QIAamp Fast DNA Stool Mini Kit (Qiagen, Hilden, Germany) according to the manufacturer’s instructions. The V3-V4 region of 16S rRNA genes were amplified by forward primer 5′-ACT CCT ACG GGA GGC AGC AG-3′ and reverse primer 5′-GGA CTA CHV GGG TWT CTA AT-3′. The amplified products were pooled and purified using Ampure magnetic purification beads (Agencourt, CA, USA). After library construction, the amplicons were paired-end sequenced (2 × 250) on a MiSeq platform (Illumina, San Diego, CA, USA) at the Beijing Genomics Institute (BGI, Beijing, China). 

### 2.12. Bacterial Quantification in Feces

For quantification of total fecal bacterial load, total DNA was isolated from known amounts of feces using QIAamp DNA Stool Mini Kit (Qiagen, Hilden, Germany). DNA was then subjected to quantitative PCR using QuantiFast SYBR Green PCR kit (Qiagen, Hilden, Germany) with universal 16S rRNA primers (Appendix A) to measure the total bacterial count. Results are expressed as bacterial count per mg of stool, using a standard curve.

### 2.13. Statistical Analysis

Data were expressed as mean ± SEM and multiple groups were compared using repeated-measures ANOVA plus Tukey’s test (SAS 9.2 SAS Institute Inc., Cary, NC, USA). In case of skewed values, the data were logarithmically transformed to normalize their distribution. *p* values ≤ 0.05 were considered statistically significant.

## 3. Results

### 3.1. CSF Promoted Weight Loss during the NCD But Not HFD Feeding Period

The DIO mice were fed HFD supplemented with 0%, 4%, or 6% CSF and lean mice were fed LFD for 12 weeks (period 1) The respective diets were substituted with normal chow or adjusted chow for another 4 weeks (period 2) (see Section 2 and Appendix A). As shown in Figure 1A, CSF supplementation had no effect on the adiposity status of the HFD-fed mice, but markedly decreased their body weight compared to the HF-C group once the animals were put on NCD. In addition, supplementing NCD with 4% or 6% CSF also decreased Lee’s index (Appendix A), white adipose tissue weight (Figure 1B), adipocyte size (Figure 1C and Appendix A) and hepatic lipid content (Figure 1D,E), without affecting the liver weight (Appendix A). While the serum lipid profile was not influenced by 6% CSF supplementation during period 1, the levels of low-density lipoprotein cholesterol (LDL-C) and high-density lipoprotein cholesterol (HDL-C) were respectively decreased and increased during period 2 compared to those in the HF-C group (Appendix A). Furthermore, CSF did not influence energy intake during period 1 but significantly attenuated both energy intake during period 2 (Figure 1F). In the 24 h feeding pattern test during period 2, 6% CSF decreased food intake between 4 and 24 h as well as the entire day compared to the HF-C group (Appendix A), indicating that high dose CSF supplementation enhanced satiety. Consistent with this, both 4% and 6% CSF increased the levels of PYY and GLP-1 during period 2 compared to those in the HF-C group (Figure 1G). Taken together, CSF supplementation significantly reduced the adiposity of obese mice only when administered with NCD.

### 3.2. CSF Alleviated Obesity-Related Insulin Resistance

HFD markedly increased glucose intolerance and insulin resistance, as indicated by the greater area under curve (AUC) of glucose both in the intraperitoneal glucose tolerance test (IGTT) and intraperitoneal insulin tolerance test (IITT) (Figure 2C,D). Supplementing HFD with 6% CSF decreased the 6 h fasting blood glucose level with a tendency (period 1) (Figure 2A), and alleviated the insulin resistance with a decreased AUC of glucose in IITT (Figure 2D). Switching to NCD, the discrepancies in glucose intolerance and insulin resistance between LF-C and HF-C groups disappeared (Figure 2E,F). Supplementing NCD with 6% CSF significantly decreased the 6 h fasting blood glucose level (Figure 2A), and maintained an increased insulin sensitivity (Figure 2D). In addition, 4% CSF supplementation showed an improvement in insulin sensitivity during period 2 (Figure 2F). There wsa no difference in the fasting insulin levels among all groups during either feeding period (Figure 2B). Taken together, 6% CSF supplementation improved insulin sensitivity in HFD-induced obese mice as well as weight-losing mice.

### 3.3. CSF Attenuated Systemic Inflammation in Obese Mice by Enhancing Intestinal Morphological Integrity

Insulin resistance is closely related to obesity-related systemic inflammation [32]. Consistent with this, HFD feeding significantly increased the serum levels of LPS, IL-6, tumor necrosis factor-α (TNF-α), and leptin compared to that in the LFD-fed mice (Figure 3A). However, 6% CSF supplementation decreased the levels of these inflammatory markers in the obese mice compared to the HF-C group (Figure 3A). Furthermore, both 4% and 6% CSF markedly decreased the circulating leptin levels and increased that of adiponectin, an adipocyte-secreted endogenous insulin sensitizer (Figure 3A). The assuaged inflammatory status might correlate with increased morphological integrity of colon tissue in the CSF-fed obese mice, as indicated by increased colonic mucosal layer thickness and crypt length in the 6% CSF group compared to that in the HF-C group (Figure 3C,D). Such promotion of colon tissue morphology by CSF correlated with increased enterocyte proliferation, as measured by the number of BrdU-positive cells in the proximal colon section (Figure 3E,F). In addition, 6% CSF supplementation upregulated mRNA expression of tight junction gene *ZO-1* and downregulated mRNA expression of pore-forming protein gene *Claudin-2* (Figure 3B). Taken together, 6% CSF attenuated obesity-induced systemic inflammation and enhanced the gut morphological integrity.

### 3.4. CSF Modulated the Intestinal Microbiota

The gut microbiota of the differentially-treated mice was analyzed by deep sequencing the 16S rRNA V3-V4 region in fecal samples. The total bacteria load of feces was also measured by quantitative PCR. As shown in Figure 4A, HFD and CSF-supplemented HFD did not influence total bacteria load of mice compared that of LFD-fed mice during period 1, while 6% CSF supplementation significantly increased the total bacterial load compared to that of mice from HF-C and LF-C groups during period 2. There were no differences in the alpha diversity (Shannon index and observed operational taxonomic units (OTUs)) among the four groups during either feeding period (Figure 4B and Appendix A). However, substituting the HFD with NCD significantly reduced the alpha diversity. In contrast, the beta diversity index (PCA and unweighted UniFrac distance) was significantly different in the HFD- and the LFD-fed mice during period 1, indicating greater dissimilarity in the microbiota composition of the former, which, however, decreased during period 2 (Figure 4C–E). The clustering of the 6% CSF group showed a tendency to deviate from that of HF-C group during period 1, which increased further during period 2 (Figure 4C). Intriguingly, 6% CSF markedly increased the unweighted UniFrac distance from the LF-C group during period 2 (Figure 4E), which was inconsistent with the PCA result and might be related to the increase in intra-group UniFrac distance (Figure 4D). At the phylum level, HFD significantly increased the relative abundance of Proteobacteria and reduced that of Bacteroidetes (Figure 4F,H) and increased the Firmicutes/Bacteroidetes (F/B) ratio (Figure 4G). After switching to NCD, the F/B ratio decreased in the HF-C group, whereas 6% CSF supplementation resulted in a sequential decrease in Firmicutes and increase in Bacteroidetes. Furthermore, 4% and 6% CSF also decreased the relative abundance of Proteobacteria compared to the HF-C group during period 2.

### 3.5. CSF Regulated the Production of SCFAs

The levels of acetate, propionate, butyrate, valerate, isobutyrate, and isovalerate in the feces were analyzed by gas chromatography. As shown in Figure 5A, the fecal levels of SCFAs increased significantly in the obese versus lean mice during period 1, and CSF supplementation in the former restored the levels of all but acetate in a dose-dependent manner. The SCFAs decreased significantly in the HF-C mice to levels similar to that in mice from LF-C group during period 2 (Figure 5B). In contrast, 6% CSF maintained similar levels of SCFAs throughout the 16 week period and increased the levels of acetate, propionate and total SCFAs compared to the mice from HF-C group during period 2 (Figure 5B). Taken together, 6% CSF supplementation stabilized SCFA production in the intestine of DIO mice.

### 3.6. Correlation between Gut Microbiota and the Metabolic Parameters

Clustering of the fecal microbiota at the genus level revealed three clusters corresponding to the pre-treatment, and feeding periods 1 and 2 (Figure 6A). Furthermore, metastats analysis indicated that 14 genera were increased and six were decreased in the HF-C group compared to the LF-C group during period 1 (Figure 6B). Specifically, six genera belonging to class Clostridiales phylum Firmicutes (*Dorea*, *Ruminococcus*, *Coprococcus*, *Roseburia*, *Blautia*, and *Oscillospira*), *Staphylococcus* (class Bacillales phylum Firmicutes), *Butyricimonas* (phylum Bacteroidetes), and six unclassified genera from class Clostridiales and families Christensenellaceae, Lachnospiraceae, Erysipelotrichaceae, Mogibacteriaceae and Desulfovibrionaceae showed increased abundance in the HFD-fed mice. Most of the above have the ability to produce SCFAs. After switching to NCD, two genera (*Paraprevotella* and *Desulfovibrio*) increased, and four genera (*Natronobacillus*, *Staphylococcus*, *Dorea*, and *Cupriavidus*) decreased in the HF-C, compared to LF-C-fed mice. Supplementation with 6% CSF increased the relative abundance of *Aerococcus* and decreased that of *Cupriavidus*, *Ochrobactrum* and unclassified_Clostridiaceae in the HFD-fed mice during period 1 (Figure 6C). During period 2, 6% CSF increased the abundance of *Allobaculum*, *Clostridium* (family Erysipelotrichaceae phylum Firmicutes) and three unclassified genera (unclassified S24-7, unclassified Coriobacteriaceae and unclassified RF39), and decreased that of *Roseburia*, *Ruminococcus*, *Coprococcus*, *Oscillospira*, *Dehalobacterium* (class Clostridiales phylum Firmicutes), *Leuconostoc*, *Streptococcus* (class Lactobacillales phylum Firmicutes), *Desulfovibrio* (phylum Proteobacteria), and seven unclassified genera from class Clostridiales and families Lachnospiraceae, Ruminococcaceae, Mogibacteriaceae, Desulfovibrionaceae, and Bifidobacteriaceae. Spearman analysis further showed that the high abundance of *Allobaculum* and Unclassified_S24-7 in the 6% CSF group were negatively correlated with body weight, weight gain, and insulin resistance (Figure 6D), and the decreased abundance of *Roseburia*, *Ruminococcus*, *Coprococcus*, *Oscillospira*, *Dehalobacterium*, *Streptococcus*, Unclassified Clostridiales, Unclassified Lachnospiraceae, Unclassified Ruminococcaceae, Unclassified Christensenellaceae, and Unclassified Desulfovibrionaceae was positively correlated with body mass and insulin resistance. In addition, *Oscillospira*, Unclassified Clostridiales, Unclassified Lachnospiraceae, Unclassified Ruminococcaceae, and Unclassified Desulfovibrionaceae had the highest number of edges and positive correlations in the co-occurrence network analysis (r ≥ 0.7 and *p* ≤ 0.05) (Figure 6E), indicating a key role in the microbial response to CSF. Taken together, CSF-mediated changes in the intestinal microbiota were closely related to the improvements in host obesity and insulin sensitivity.

## 4. Discussion

Prolonged caloric surplus is the primary cause of obesity [33,34]. Dietary fiber regulates metabolism on account of its physicochemical properties [35,36,37]. Viscous fibers have been associated with altered blood glucose and cholesterol levels, prolonged gastric emptying, and slower transit through the small intestine [36]. Fermented fibers produce SCFAs that promote glucose homeostasis [37]. The physiological responses to viscous and fermented dietary fibers eventually promote and prolong satiety. In a previous study, we found that supplementing rat feed with 2% CSF reduced food intake by promoting satiety [30,31]. In the present study as well, CSF significantly promoted satiety and increased weight loss when HFD was replaced with the normal feed. However, CSF did not affect the adiposity status or energy intake of DIO mice during HFD feeding, indicating that high levels of dietary fat nullifies the feelings of satiety induced by CSF. Dipatrizio and Piomelli (2015) reported that the ingestion of fat-rich foods initiates a cephalically-driven induction of endocannabinoid activity in the gut and maximizes food consumption [38]. Rossi et al. (2019) reported that HFD weakened the activity of glutamatergic nerve cells in the lateral hypothalamus, which promoted overeating and obesity [39]. Furthermore, leptin resistance in obese individuals is a potential cause of eating disorders [40,41]. Leptin is an adipocyte-derived pleiotropic molecule that regulates food intake as well as metabolic and endocrine functions, immunity, inflammation, and hematopoiesis [42]. The most important role of leptin is to signal the brain regarding nutrient availability and modulate energy intake accordingly. Interestingly, most obese individuals have high levels of circulating leptin, which is seemingly inconsistent with obesity, indicating a state of leptin resistance [43]. We detected higher leptin levels in the sera of HFD-fed mice, which was decreased by CSF supplementation.

Insulin resistance is a major driver of many chronic metabolic disorders. There are many potential molecular causes of insulin resistance; ultimately, they are all either directly or indirectly caused by increased inflammation [32]. Systemic inflammation damages the pancreatic beta cells, disrupts insulin action and mediates glucose intolerance in obese individuals [11], and LPS-derived chronic low-grade inflammation is one of the hallmarks of obesity [11,44]. Furthermore, high levels of circulating LPS are associated with increased intestinal permeability [11]. In the present study, 6% CSF supplementation improved the insulin sensitivity of DIO mice during both feeding periods, enhanced intestinal morphological integrity, and decreased circulating LPS levels and inflammation induced by the HFD.

The DIO mice had a significantly lower relative abundance of Bacteroidetes, increased abundance of Proteobacteria and higher F/B ratio compared to the lean mice, which is consistent to previous reports [24,45]. Furthermore, the obese mice showed significantly greater abundance of SCFA-producing bacteria of phylum Firmicutes, along with higher fecal levels of the SCFAs. Previous studies have shown that the gut microbiota of obese donors produce excessive amounts of SCFAs that increase colonic energy availability and exacerbate weight gain of mice recipients [10]. It is confirmed in human that obese individuals have higher fecal levels of SCFAs compared with that of lean individuals without difference in SCFA absorption, which is strongly positively correlated to Phylum Firmicutes [46]. However, some studies in mice report that HFD have a slight or even reduced effect on fecal SCFAs compared to normal diet [47,48]. This discrepancy might be related to the differences in diet formulation, treatment duration, and intestinal microbiota composition as well as SCFA absorption. In the current study, the aberrant production of SCFAs might affect the intestinal barrier function as well. SCFAs are the major products of microbial fermentation [49] of undigested dietary carbohydrates, host-derived glycans (mainly mucins), and, to a lesser extent, proteins and peptides [50,51]. When fermentable fibers are in short supply, the gut microbes alter their energy source to host glycans, such as the glycoprotein-rich mucus layer [52,53], which is accompanied by an expansion of mucus-degrading bacteria [51]. Furthermore, we detected high levels of BCFAs (isobutyrate and isovalerate), the primary end products of protein fermentation [54], in the HFD-fed mice, indicating greater microbial fermentation of proteins and peptides. Protein degradation by the gut bacteria generates harmful metabolites like ammonia, phenols, and hydrogen sulfide, which can damage the colonic epithelium structure [55]. In addition, excessive protein fermentation can also deplete the proteins in the mucus layer. Therefore, HFD-induced alterations in the gut microbiota disrupt the intestinal barrier likely via excessive protein fermentation and SCFAs production. During the HFD feeding period, 6% CSF supplementation significantly decreased the relative abundance of Firmicutes as well as the F/B ratio. Consistent with the altered microbiota, 6% CSF supplementation significantly decreased the fecal levels of propionate, butyrate, valerate and BCFAs. CSF continued to reduce Firmicutes and increase Bacteroidetes abundance in the obese mice after HFD was replaced with NCD, in addition to decreasing the relative abundance of Proteobacteria and increasing that of Actinobacteria. Furthermore, 6% CSF supplementation significantly increased the total bacteria load, which was beneficial for the maintenance of the ecosystem stability [56]. The fecal levels of SCFAs were also consistent during the entire 16 week feeding period in the 6% CSF group. Thus, an adequate amount of CSF can stabilize the microbiota and intestinal SCFAs production and then likely restore intestinal morphological integrity as well, which could be related to its function as a carbohydrate substrate.

While the genus composition of the gut microbiome was only marginally affected by CSF during the HFD feeding period, it was sharply altered upon NCD substitution, which paralleled the changes in the obesity phenotype. CSF significantly increased the abundance of *Allobaculum* and Unclassified_S24-7 (S24-7 family) during the NCD feeding period. *Allobaculum* is a beneficial intestinal bacterium [57,58] that produces lactic and butyric acids and small amounts of ethanol from glucose [57] and decreases glucose digestion [59]. Several studies have correlated increased abundance of *Allobaculum* with a lean phenotype [60] and glucose tolerance [61]. S24-7, a dominant family belonging to the class Bacteroidales, degrades complex polysaccharides into acetate, propionate, and succinate [62]. We also detected a negative correlation of *Allobaculum* and Unclassified_S24-7 abundance with obesity and insulin resistance. Furthermore, the 11 genera decreased by CSF supplementation during the NCD period were positively correlated with obesity and insulin resistance. These results point to the possibility that CSF mitigates obesity and insulin sensitivity by regulating the gut microbiota and decreasing energy intake. *Oscillospira*, Unclassified Clostridiales, Unclassified Lachnospiraceae, Unclassified Ruminococcaceae, and Unclassified Desulfovibrionaceae were decreased by CSF and showed the highest number of positive correlations, indicating that they likely play a key role in the response of the intestinal microbiota to CSF. It is worth noting that *Oscillospira* species are more likely to degrade sugars released from host mucins [63,64], which supports the hypothesis that CSF prevents mucin depletion.

## 5. Conclusions

In summary, the SCFA-producing bacteria of phylum Firmicutes were predominant in the gut of the HFD-fed obese mice, which might excessively consume host-derived substrates, and then impair the intestinal barrier, aggravate systemic inflammation, and promote insulin resistance. Supplementing HFD with 6% CSF stabilized gut microbiota composition, and improved insulin sensitivity of obese mice, although no impact on obesity phenotype was observed. After switching to NCD, 6% CSF supplementation significantly enhanced the satiety and promoted weight loss. In addition, the stabilizing effects of CSF were more prominent, as indicated by a lower F/B ratio, greater total bacterial load, and maintained SCFAs production. The intestinal microbiota homeostasis and stabilized the production of short-chain fatty acids by CSF led to an increased intestinal morphological integrity and decreased systemic inflammation, which might also be beneficial for obesity control.

## Figures and Tables

**Figure 1 nutrients-12-00351-f001:**
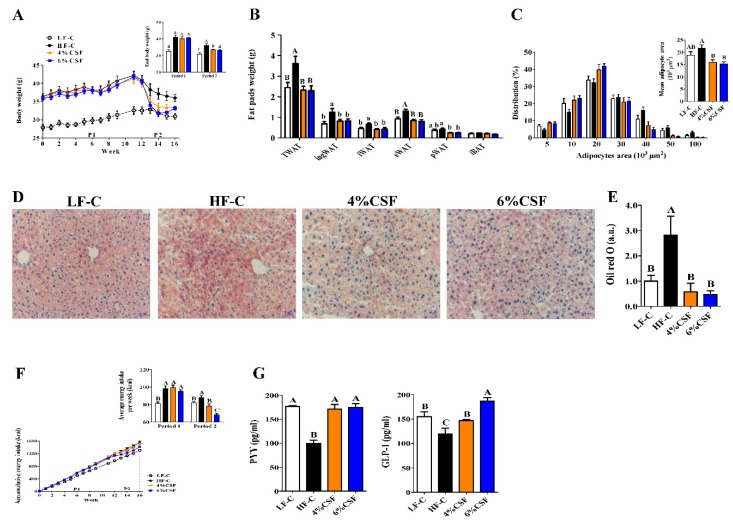
Combined soluble fiber (CSF) improves the adiposity in diet-induced obese (DIO) mice by regulating appetite under normal chow basal diet (NCD) but not high-fat diet (HFD). (**A**) Body mass over time and final body weight per period (n = 7). (**B**) Weight of total white adipose tissue (TWAT), inguinal white adipose tissue (ingWAT), interscapular WAT (iWAT), epididymal WAT (eWAT), perirenal fat (pWAT) and interscapular brown adipose tissue (iBAT) (n = 6–7). (**C**) Adipocyte size distribution (n = 7). (**D**,**E**) Fat in the liver (n = 6–7) was observed by oil red staining (**D**) and quantified by image analysis at 200 × magnification (**E**). (**F**) Accumulative energy intake (n = 7). (**G**) PYY and GLP-1 levels in the 2-h postprandial blood during period 2 (n = 6–7). Data are expressed as mean ± SEM. Statistical significance was assessed by Tukey’s test for multiple comparisons. ab means in the same bar without a common letter differ at *p* < 0.05; ABC means in the same bar without a common letter differ at *p* < 0.01.

**Figure 2 nutrients-12-00351-f002:**
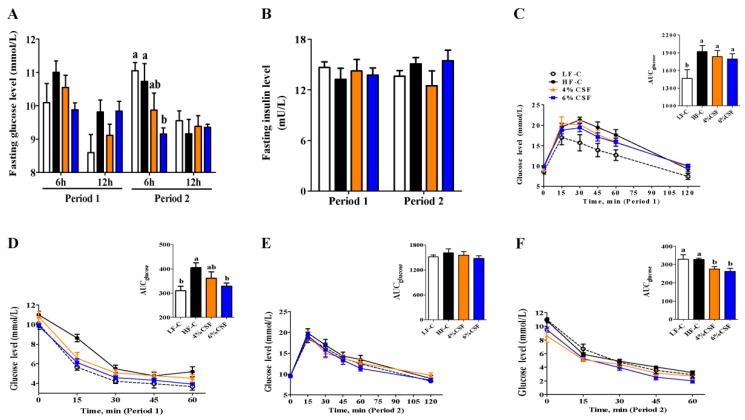
CSF improves insulin sensitivity in DIO Mice. (**A**) 6-and 12-h fasting serum glucose at the end of period 1 and period 2 (n = 6–7). (**B**) Overnight fasting insulin levels at the end of period 1 and period 2 (n = 6). (**C**, **E**) Intraperitoneal glucose tolerance test (IGTT, 2 g/kg) (n = 6–7) and the area under the curve (AUC) during period 1 (**C**) and period 2 (**E**). (**D**, **F**) Intraperitoneal insulin tolerance test (IITT, 1 IU/kg) (n = 6–7) and the AUC during period 1 (**D**) and period 2 (**F**). Data are expressed as mean ± SEM. Statistical significance was assessed by Tukey’s test for multiple comparisons. ab means in the same bar without a common letter differ at *p* < 0.05.

**Figure 3 nutrients-12-00351-f003:**
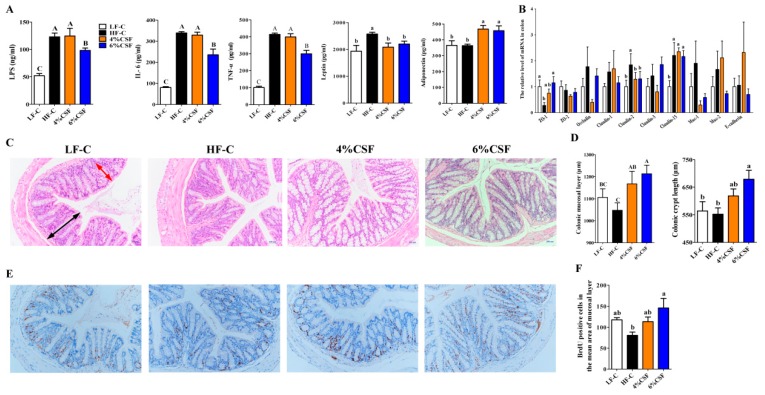
CSF improves the systemic inflammation and gut integrity of DIO mice. (**A**) lipopolysaccharide (LPS), interleukin-6 (IL-6), Tumor Necrosis Factor (TNF-α), leptin and adiponectin (n = 6). (**B**) *ZO-1*, *Occludin*, and *Claudin-2* mRNA expression levels in colon tissue (n = 6–7). (**C**) Representative H&E stained images of colon tissue at 200 × magnification. Scale bars, 200 µm. (**D**) Quantifications of crypt length and mucosal layer thickness in colon tissue (n = 7). (**E**) Representative BrdU stained proximal colon tissue and (**F**) BrdU-positive crypt cells (n = 7). Data are expressed as mean ± SEM. Statistical significance was assessed by Tukey’s test for multiple comparisons. ab means in the same bar without a common letter differ at *p* < 0.05; ABC means in the same bar without a common letter differ at *p* < 0.01.

**Figure 4 nutrients-12-00351-f004:**
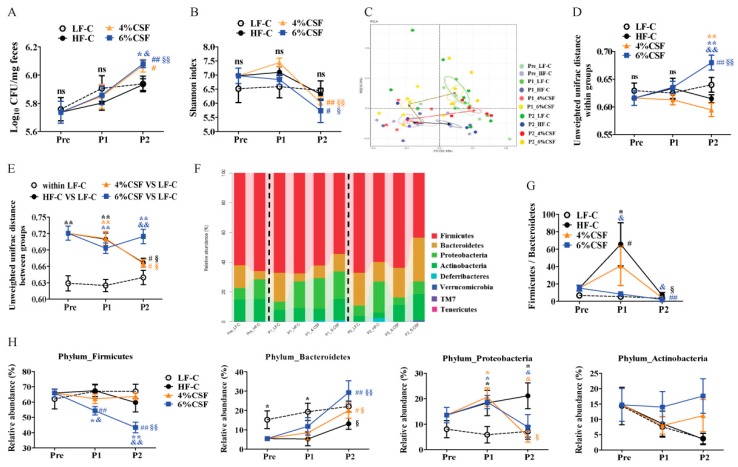
CSF alters the gut microbiota composition. (**A**) Total bacteria load (n = 7). (**B**) The alpha diversity of feces microbiota as a function of time (n = 5–7). (**C**) The principal component analysis (PCA) of feces microbiota (n = 5–7). The circle shows a 10% confidence interval (CI). (**D**,**E**) The unweighted unifrac distance within (**D**) and between (**E**) groups calculated at the operational taxonomic unit (OTU) level across individual microbiota. (**F**) The changes in bacterial abundance at the phylum level. (**G**) The Firmicutes/Bacteroidetes ratio in different groups. (**H**) The changes in relative abundance of Firmicutes, Bacteroidetes, Proteobacteria and Actinobacteria. Data are expressed as mean ± SEM. Statistical significance was assessed by Tukey’s test for multiple comparisons. *, ** *p* < 0.05 and 0.01 respectively compared with the LF-C group. & and && indicate *p* < 0.05 and 0.01, respectively, compared with the HF-C group. # and ## indicate *p* < 0.05 and 0.01, respectively, compared with the pre-treatment group. § and §§ inicate *p* < 0.05 and 0.01, respectively, compared with the treatment period 1. ns —not significant, *p* > 0.05. The black, orange, and blue symbols represent the statistical analysis of HF-C, 4% CSF, or 6% CSF groups, respectively, with other groups or the same group at different time points.

**Figure 5 nutrients-12-00351-f005:**
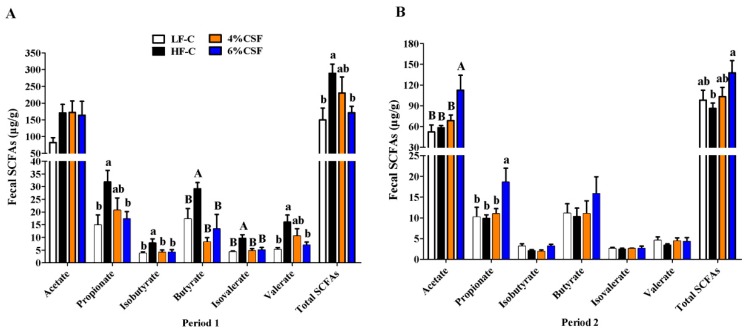
Effect of CSF supplementation on the production of short-chain fatty acids (SCFAs) during the two consecutive periods. Fecal SCFAs (n = 5–7) as measured by gas chromatography during period 1 (**A**) and period 2 (**B**). Data are expressed as mean ± SEM. Statistical significance was assessed by Tukey’s test for multiple comparisons. ab means in the same bar without a common letter differ at *p* < 0.05; AB means in the same bar without a common letter differ at *p* < 0.01.

**Figure 6 nutrients-12-00351-f006:**
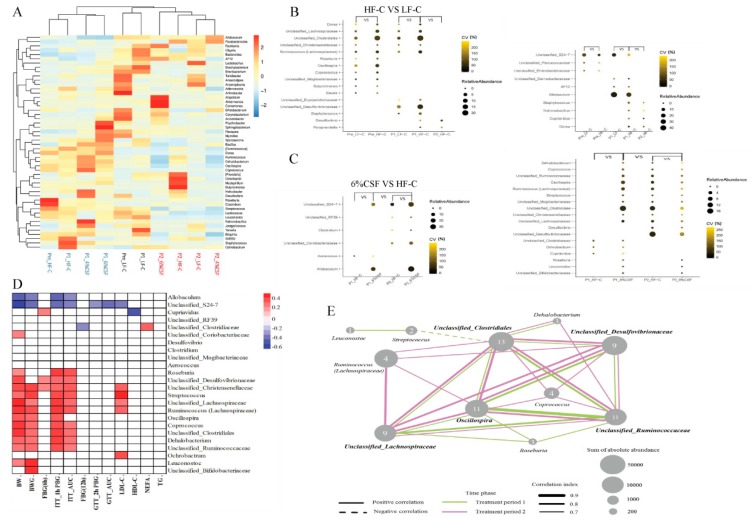
Microbial taxonomy and interaction of CSF on fecal microbiota over time. (**A**) Clustering of the dental plaque microbiota at the genus level. (**B**) Significantly differential genera in HFD relative to LFD-fed mice. (**C**) Significantly differential genera at each time point in 6% CSF group with the horizontal and longitudinal comparisons by Metastats analysis. The sizes and colors of the circles indicate the average abundance and the coefficient of variance (CV) of the abundance, respectively. (**D**) Correlation of differential genera with obesity and glycolipid metabolism. BW—body weight; BWG—body weight gain; FBG(6 h) —6 h fasting blood glucose; ITT_1h PBG—blood glucose at 1 h post-injection in insulin tolerance test (ITT); ITT_AUC—area under curve of glucose in ITT; FBG(12 h) —12 h fasting blood glucose; GTT_1h PBG—blood glucose at 2 h post-injection in glucose tolerance test (GTT); GTT_AUC—area under curve of glucose in GTT; LDL-C—low-density lipoprotein-cholesterol; HDL-C—high-density lipoprotein-cholesterol; NEFA—non-esterified fatty acid; TG triglyceride. (**E**) Co-occurrence network of the significantly differential genera. The thresholds of SparCC correlations were r ≥ 0.7 and *p* ≤ 0.05. The pink and green lines represent the connection between two genera during treatment period 1 and period 2, respectively. The dotted and solid lines indicate negative correlation and positive correlation, respectively. The thickness of the line is proportional to correlation value. The size of the nodes is proportional to their relative abundance, and the number marked on each node represents the degree of this node.

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
