# Peer review of "Combined Soluble Fiber-Mediated Intestinal Microbiota Improve Insulin Sensitivity of Obese Mice"

_nutrients, 2020, doi:10.3390/nu12020351_

Round 1
Reviewer 1 Report
Authors used t-test for all data analysis, which is not correct method for analyzing data set that contains more than two groups. Please re-analyze all results by the appropriate statistical method.Authors should provide more background information about the combined soluble fiber (CSF). What is the composition of the CSF? What are the advantages of CSF compared to other types of soluble fiber? The fact that soluble fiber can alter microbiota and improve metabolic syndrome is not new.
In figure 3C, I cannot tell increased colonic mucosal layer thickness and crypt length in the 6% CSF group.
Authors stated CSF improved insulin sensitivity in DIO mice. In my opinion, the change is quite minor. Authors should soft the conclusion.
To test barrier integrity, authors should perform Fitc-dextran experiment. Only mRNA of ZO-1 is upregulated in CSF groups, but not other genes. Author should investigate more barrier function genes, or test those at protein level by western blot.
Previous studies have demonstrated that high fat diet can reduce SCFA formation, whereas supplementary soluble fiber increases SCFA production (PMID: 24236183; 30340040). Authors presented opposite results.
What is the total gut bacteria load?
The overall writing is poor. Extensive editing is required.
Minor:
C57BL/6J means C57BL/6 mice from Jackson Laboratory. Obviously, the mice used in this manuscript is not from Jackson. To use an acronym (GLP-1 and PYY), present both full form and acronym for the first time. Method of measuring Lee’s index is missing. “slaughter” is not an appropriate word to use. “Slaughter” means kill for food.Author Response
RE: Combined Soluble Fiber-mediated Intestinal Microbiota Improve Insulin Sensitivity Of Obese Mice (nutrients-679931)
Dear reviewer,
Thank you very much for your warm composition comments concerning our manuscript entitled "Combined Soluble Fiber-mediated Intestinal Microbiota Improve Insulin Sensitivity Of Obese Mice" (MS ID: nutrients-679931).
Here we submit a new version of our manuscript, which has been modified accordingly to the comments. Detailed point-by-point responses are provided below. We mark all the changes by using red font in the revised manuscript.
We want to do our best to meet the requirements. If you have any question about this paper, please don’t hesitate to let us know.
Sincerely,
Rong Zheng and Chuanhui Xu
Comment 1: Authors used t-test for all data analysis, which is not correct method for analyzing data set that contains more than two groups. Please re-analyze all results by the appropriate statistical method.
Response: Thanks for your comments. We have reanalyzed all results with the repeated ANOVA measures plus Tukey’s test for multiple comparisons (SAS 9.2 SAS Institute Inc., Cary, NC, USA). Please see lines 178-179 of the revised manuscript. Minor statistical differences existed and are modified in the revised manuscript.
Comment 2: Authors should provide more background information about the combined soluble fiber (CSF). What is the composition of the CSF? What are the advantages of CSF compared to other types of soluble fiber? The fact that soluble fiber can alter microbiota and improve metabolic syndrome is not new.
Response: Thanks for your suggestion. We have provided relevant information about CSF. Please see lines 65-73 of the revised manuscript. The composition and proportion of CSF is provided in Animal diets section, please see lines 84-86 of the revised manuscript. The combination of dietary fibers synergistically enhances physiochemical properties and physiological functions, which increases the choice of fiber sources in the diet. Compared with other types of soluble fiber, the main ingredient of CSF have greater annual output and low prices, which promoted the utilization of dietary fiber in other yields, such as animal production.
Comment 3: In figure 3C, I cannot tell increased colonic mucosal layer thickness and crypt length in the 6% CSF group.
Response: Thanks for your comment. 7 colonic tissue samples per group were stained with hematoxylin and eosin for histologic observation. After checking, pictures of 4% and 6% CSF groups could not represent the average of whole group. Representative images have been selected in the revised manuscript.
Comment 4: Authors stated CSF improved insulin sensitivity in DIO mice. In my opinion, the change is quite minor. Authors should soft the conclusion.
Response: Thanks for your comment. In the present study, high-fat diet (HFD) markedly increased glucose intolerance and insulin resistance, as indicated by 31.11% and 30.38% increases in AUCs of glucose in the IGTT and IITT, respectively (Fig. 2C and D). Supplementing HFD with 6% CSF enhanced the hypoglycemic effect of exogenous insulin in obese mice, as indicated by a 18.70% decrease in AUC of glucose in IITT compared to that of HF-C group (Fig. 2D), while 6% CSF supplementation did not effect the ability of obese mice in processing exogenous glucose (Fig. 2C). Switching to normal chow diet (NCD), 6% CSF supplementation kept the similar effects on IGTT and IITT, as indicated by a 19.76% decrease in AUC of glucose in IITT (Fig. 2F) and no difference in AUC in IGTT compared to that of HF-C group (Fig. 2E). The response to exogenous glucose in vivo are complexly regulated by many factors, such as blood glucose sensing, insulin releasing, the amount of insulin released as well as the sensitivity of insulin receptor, which might influence the effect of CSF on exogenous glucose in IGTT. In conclusion, 6% CSF improved the insulin sensitivity, but slightly effected the glucose intolerance. We have soft the conclusion, please see lines 222-223 of the revised manuscript.
Comment 5: To test barrier integrity, authors should perform Fitc-dextran experiment. Only mRNA of ZO-1 is upregulated in CSF groups, but not other genes. Author should investigate more barrier function genes, or test those at protein level by western blot.
Response: Thanks for your comment. It was a pity that Fitc-dextran experiment was not carried out. In the current study, we assessed the morphological changes of colon tissue by histological H&E stain, and found that 6%CSF supplementation promoted enterocyte proliferation, and then improved intestinal morphological integrity, as indicated by the increased BrdU-positive cells in crypt (Fig. 3E and F), mucosal layer thickness and crypt length (Fig. 3C and D). For assessing the intercellular permeability, we have tested more genes related to tight junction function. We found that HFD significantly downregulated mRNA expression of tight junction gene ZO-1, and upregulated mRNA expression of pore-forming protein gene Claudin-2 and Claudin-15 (Fig. 3B). Claudin-2 and Claudin-15 promote the formation of paracellular ion pores [1]. In previous study, expression of Claudin-2 is highly upregulated during inflammatory bowel disease (IBD) and, due to its association with epithelial permeability, has been postulated to promote inflammation [2,3]. Claudin-15 also have been reported to form a “leaky” barrier in certain tissues [4]. 6% CSF supplementation upregulated mRNA expression of ZO-1 gene and downregulated the mRNA expression of Claudin-2 gene in colon (Fig. 3B). Although few genes were regulated by CSF, it also might be important for the maintain of the intestinal barrier. Thanks for your suggestion again. We have revised our conclusion to that 6% CSF supplementation improved the intestinal morphological integrity, please see lines 245-247 of the revised manuscript.
Comment 6: Previous studies have demonstrated that high fat diet can reduce SCFA formation, whereas supplementary soluble fiber increases SCFA production (PMID: 24236183; 30340040). Authors presented opposite results.
Response: Thanks for your comment. The formation of SCFAs is closely related to the substrates and microbial composition. Gut microbiota is a large and complex ecosystem with great individual differences. Different diet causes multidirectional changes in the gut microbiota. High-fat, high-carbohydrate diet might differently effect the composition and metabolic activity of gut microbiota compared with high-fat, low-carbohydrate, although both of them contain high level of dietary fat. In the present study, the high-fat, low-fiber diet significantly promoted an expansion of SCFAs-producing bacteria belonged to Firmicutes along with a higher formation of faecal SCFAs, which was consistent with previous reports [5-7]. 6% CSF supplementation stabilized microbiota composition with decreasing relative abundance of Firmicutes and increasing relative abundance of Bacteroidetes, and stabilizing the production of SCFAs during high-fat basal diet period and normal chow diet period. As to the differences between the present study and previous studies (PMID: 24236183; 30340040) which supported high-fat diet can reduce SCFA formation and soluble fiber increases SCFA production, it might be related to the differences in high-fat diet formulation, dietary treatment duration as well as individual difference of gut microbiota. Therefore, more research is necessary.
Comment 7: What is the total gut bacteria load?
Response: Thanks for your comment. We have tested the total gut bacteria load. Result showed that HFD and CSF-supplemented HFD did not influence total bacteria load of mice compared that of low-fat diet (LFD)-fed mice during period 1, while 6% CSF supplementation significantly increased the total bacterial load compared to that of mice from HF-C and LF-C groups during period 2 (Fig. 4A).
Comment 8: The overall writing is poor. Extensive editing is required.
Response: Thanks for your suggestion. We have employed the services of a professional proof-reader to improve the use of English.
Comment 9: C57BL/6J means C57BL/6 mice from Jackson Laboratory. Obviously, the mice used in this manuscript is not from Jackson. To use an acronym (GLP-1 and PYY), present both full form and acronym for the first time. Method of measuring Lee’s index is missing. “slaughter” is not an appropriate word to use. “Slaughter” means kill for food.
Response: Thanks for your comments. We have corrected “C57BL/6J” to “ C57BL/6” (line 89 of revised manuscript), and present both full form and acronym of GLP-1 and PYY for the first time (lines 61-62 of revised manuscript). We have supplied the method of measuring Lee’s index (lines 9-10 of the revised supplementary materials). “at slaughter” is replaced to “at termination of study” (line 9 of the revised supplementary materials).
Krause, G.; Winkler, L.; Mueller, S.L.; Haseloff, R.F.; Piontek, J.; Blasig, I.E. Structure and function of claudins. Biochimica et Biophysica Acta (BBA)-Biomembranes 2008, 1778, 631-645. Weber, C.R.; Nalle, S.C.; Tretiakova, M.; Rubin, D.T.; Turner, J.R. Claudin-1 and claudin-2 expression is elevated in inflammatory bowel disease and may contribute to early neoplastic transformation. Laboratory investigation 2008, 88, 1110. Zeissig, S.; Bürgel, N.; Günzel, D.; Richter, J.; Mankertz, J.; Wahnschaffe, U.; Kroesen, A.J.; Zeitz, M.; Fromm, M.; Schulzke, J.D. Changes in expression and distribution of claudin 2, 5 and 8 lead to discontinuous tight junctions and barrier dysfunction in active Crohn’s disease. Gut 2007, 56, 61-72. Colegio, O.R.; Van Itallie, C.M.; McCrea, H.J.; Rahner, C.; Anderson, J.M. Claudins create charge-selective channels in the paracellular pathway between epithelial cells. American Journal of Physiology-Cell Physiology 2002, 283, C142-C147. Riva, A.; Borgo, F.; Lassandro, C.; Verduci, E.; Morace, G.; Borghi, E.; Berry, D. Pediatric obesity is associated with an altered gut microbiota and discordant shifts in F irmicutes populations. Environmental microbiology 2017, 19, 95-105. Ley, R.E.; Bäckhed, F.; Turnbaugh, P.; Lozupone, C.A.; Knight, R.D.; Gordon, J.I. Obesity alters gut microbial ecology. Proceedings of the National Academy of Sciences 2005, 102, 11070-11075. Turnbaugh, P.J.; Ley, R.E.; Mahowald, M.A.; Magrini, V.; Mardis, E.R.; Gordon, J.I. An obesity-associated gut microbiome with increased capacity for energy harvest. nature 2006, 444, 1027.

Reviewer 2 Report
Comments to the Author
In this manuscript, the authors investigate the role of combined soluble fiber (CSF), which modulates microbiota and improves insulin sensitivity in obese mice. They suggest that in HFD- induced mice on normal chow diet (NCD), CSF 6% stabilizes the production of short-chain fatty acids, improves intestinal barrier function, systemic inflammation, improves insulin sensitivity by promoting intestinal microbiota.
This is an interesting study to show HFD replaced with NCD along with 6% CSF shows a beneficial effect on insulin sensitivity, systemic inflammation, and intestinal barrier function by altering the gut microbiota. However, there are some suggestions and concerns which should be addressed.
Major comments
What was the rationale for the concentration percentages used in this study (4% and 6 % CSF)? Furthermore, why only male mice used? Must justification be provided? Did Specific pathogen-free (SPF) population maintain until the end of the study, after the dietary interventions? Did the authors check for that? The majority of the figures are not convincing, and clarity must be improved (increase dpi); it is tough to read even in softcopy after zooming. Insert bar diagrams must be mentioned clearly in the legends along with sub-sections. The critically important thing the subpanels should not be random orders; highly distracting and hard to find in figures, it must be uniform. Check all the figures for better resolutions and rearrange them.Other:
Provide correct company name, place, and catalog for all chemicals and assay kits. What is the n's for the q RT PCR experiments (biological or technical replicates)? Mention the same for the other experiments in the legends. Some of the histology is not convincing, presented do not correspond to the quantifications shown in the bar graphs. Provide better, representative images and Scale bar must appear in figures, and in legend, it should be mentioned the magnification along with scale bar value; this should fit with all the figures contains any microscopic observations. Abstract section conclusion and overall conclusion portions not matching (Seems different, though its one of the outcomes of the study) correct it. Table S1, double-check the compositions and ingredient units that must be mentioned. Table S2 must be provided with PubMed id for those oligonucleotides. Abbreviate at first instance, check throughout the manuscript. There are many other minor errors of syntax and grammar throughout the text, which need to be fixed.Author Response
RE: Combined Soluble Fiber-mediated Intestinal Microbiota Improve Insulin Sensitivity Of Obese Mice (nutrients-679931)
Dear reviewer,
Thank you very much for your warm composition comments concerning our manuscript entitled "Combined Soluble Fiber-mediated Intestinal Microbiota Improve Insulin Sensitivity Of Obese Mice" (MS ID: nutrients-679931).
Here we submit a new version of our manuscript, which has been modified accordingly to the comments. Detailed point-by-point responses are provided below. We mark all the changes by using red font in the revised manuscript.
We want to do our best to meet the requirements. If you have any question about this paper, please don’t hesitate to let us know.
Sincerely,
Rong Zheng and Chuanhui Xu
Comment 1: What was the rationale for the concentration percentages used in this study (4% and 6 % CSF)?
Response: Thanks for your comment. In our previous works, we found that 2% CSF supplementation showed a positive effect on insulin sensitivity in pregnant sows as well as pregnant rats (data not shown). Considering the differences in basal metabolic rates among species, we referred to the method of drug dose conversion between species in medicine, and converted the supplementary ratio of CSF to 4%~6% in mice.
Comment 2: why only male mice used? Must justification be provided.
Response: Thanks for you comment. Male mice were selected for several reasons as follows: 1) Male mice are more susceptible to DIO, whereby they develop obesity sooner and to a greater extent than female mice [8-10]. 2) Sex differences in mice and rats for phenotypes of diet induced insulin resistance and glucose intolerance are more pronounced, with male mice and male rats being the most affected [11-15]. On normal rodent chow diets, male ZDF rats develop severe hyperglycaemia (~400mg/dl) and hypoinsulinaemia (~1,000pmol/l) by 4 months of age. By contrast, and rather remarkably, female ZDF rats maintain normal levels of glucose (~100mg/dl) and insulin (~5,000pmol/l) throughout their life, despite developing obesity to a similar extent as the males. 3) Sexual dimorphism in obesity complications is linked to the gonadal hormones (testosterone versus oestradiol and progesterone) and to their differential effect on fat distribution. Central adiposity, especially visceral fat, is detrimental to health, while fat accretion in the lower body in the form of subcutaneous fat might actually bestow protective effects [16]. Although the mechanisms of these depot-specific effects remain poorly defined, women and female rodents have more subcutaneous and less visceral fat than their male counterparts [17,18]. Removing the ovaries, thereby ablating endogenous oestradiol and progesterone production, masculinizes fat distribution and increases the susceptibly to diet-induced insulin resistance in female rats and mice [9,19,20]. 4) The take-home message is that sex differences are the norm rather than the exception in obesity research. Analyses of genome-wide association studies (GWAS) support this notion, highlighting a sex-specific genetic blueprint that underpins the susceptibility to obesity and diabetes mellitus [21,22]. 5) Currently, there is a lopsided reliance on using male rodents in preclinical research [23], especially in drug studies. Female rodents are explicitly avoided because. of a widespread, albeit unsubstantiated [24], assumption that the female oestrous cycle induces undesirable experimental variability; however, controlling for the oestrous cycle phases is of course important. 6) males are preferred for their more pronounced disease phenotypes.
Comment 3: Did Specific pathogen-free (SPF) population maintain until the end of the study, after the dietary interventions? Did the authors check for that?
Response: Thanks for your comment. SPF mice were purchased from Laboratory Animal Center, Huazhong agricultural university, Wuhan, People’s Republic of China (Permit No. SCXK2017-0012), and raised in SPF environment provided by Laboratory Animal Center, Huazhong agricultural university. It was a pity that we did not check whether the SPF population maintained until the end of the study.
Comment 4: The majority of the figures are not convincing, and clarity must be improved (increase dpi); it is tough to read even in softcopy after zooming. Insert bar diagrams must be mentioned clearly in the legends along with sub-sections. The critically important thing the subpanels should not be random orders; highly distracting and hard to find in figures, it must be uniform. Check all the figures for better resolutions and rearrange them.
Response: Thanks for your comments. We have improved picture clarity (at least 300 dpi) and carefully examined and corrected the legend of each subsection in all figures. We have checked subpanels in all figures and orderly arranged them for a better resolution.
Comment 5: Provide correct company name, place, and catalog for all chemicals and assay kits.
Response: Thanks for your comment. We have checked the company name, place, and catalog for all chemicals and assay kits (Table S3 of the revised supplementary materials).
Comment 6: What is the n's for the q RT PCR experiments (biological or technical replicates)? Mention the same for the other experiments in the legends.
Response: Thanks for your comments. We have provided the biological replicates of each test in the legend section of all figures.
Comment 7: Some of the histology is not convincing, presented do not correspond to the quantifications shown in the bar graphs. Provide better, representative images and Scale bar must appear in figures, and in legend, it should be mentioned the magnification along with scale bar value; this should fit with all the figures contains any microscopic observations.
Response: Thanks for your comment. We check carefully all the histological pictures and corresponding quantifications. There is a error in image selection in figure 3C. Pictures of 4% and 6% CSF groups could not represent the average of whole group. Better and representative images have been provided in the revised manuscript. The magnification and scale bar value have been noted in legend. Please see line 252 of the revised manuscript.
Comment 8: Abstract section conclusion and overall conclusion portions not matching (Seems different, though its one of the outcomes of the study) correct it.
Response: Thanks for your comment. We have corrected the overall conclusion portions accordingly. Please see lines 437-446 of the revised manuscript.
Comment 9: Table S1, double-check the compositions and ingredient units that must be mentioned. Table S2 must be provided with PubMed id for those oligonucleotides.
Response: Thanks for your comment. We have checked and revised the supplementary materials accordingly.
Reviewer Comment 11: Abbreviate at first instance, check throughout the manuscript.
Response: Thanks for your comment. We have checked throughout the manuscript about the abbreviate at first instance and corrected the inappropriate abbreviations.
Reviewer Comment 12: There are many other minor errors of syntax and grammar throughout the text, which need to be fixed.
Response: Thanks for your comment. We have employed the services of a professional proof-reader to improve the use of English.
Krause, G.; Winkler, L.; Mueller, S.L.; Haseloff, R.F.; Piontek, J.; Blasig, I.E. Structure and function of claudins. Biochimica et Biophysica Acta (BBA)-Biomembranes 2008, 1778, 631-645. Weber, C.R.; Nalle, S.C.; Tretiakova, M.; Rubin, D.T.; Turner, J.R. Claudin-1 and claudin-2 expression is elevated in inflammatory bowel disease and may contribute to early neoplastic transformation. Laboratory investigation 2008, 88, 1110. Zeissig, S.; Bürgel, N.; Günzel, D.; Richter, J.; Mankertz, J.; Wahnschaffe, U.; Kroesen, A.J.; Zeitz, M.; Fromm, M.; Schulzke, J.D. Changes in expression and distribution of claudin 2, 5 and 8 lead to discontinuous tight junctions and barrier dysfunction in active Crohn’s disease. Gut 2007, 56, 61-72. Colegio, O.R.; Van Itallie, C.M.; McCrea, H.J.; Rahner, C.; Anderson, J.M. Claudins create charge-selective channels in the paracellular pathway between epithelial cells. American Journal of Physiology-Cell Physiology 2002, 283, C142-C147. Riva, A.; Borgo, F.; Lassandro, C.; Verduci, E.; Morace, G.; Borghi, E.; Berry, D. Pediatric obesity is associated with an altered gut microbiota and discordant shifts in F irmicutes populations. Environmental microbiology 2017, 19, 95-105. Ley, R.E.; Bäckhed, F.; Turnbaugh, P.; Lozupone, C.A.; Knight, R.D.; Gordon, J.I. Obesity alters gut microbial ecology. Proceedings of the National Academy of Sciences 2005, 102, 11070-11075. Turnbaugh, P.J.; Ley, R.E.; Mahowald, M.A.; Magrini, V.; Mardis, E.R.; Gordon, J.I. An obesity-associated gut microbiome with increased capacity for energy harvest. nature 2006, 444, 1027. Hong, J.; Stubbins, R.E.; Smith, R.R.; Harvey, A.E.; Núñez, N.P. Differential susceptibility to obesity between male, female and ovariectomized female mice. Nutrition journal 2009, 8, 11. Stubbins, R.E.; Holcomb, V.B.; Hong, J.; Núñez, N.P. Estrogen modulates abdominal adiposity and protects female mice from obesity and impaired glucose tolerance. European journal of nutrition 2012, 51, 861-870. Yang, Y.; Smith Jr, D.L.; Keating, K.D.; Allison, D.B.; Nagy, T.R. Variations in body weight, food intake and body composition after long‐term high‐fat diet feeding in C57BL/6J mice. Obesity 2014, 22, 2147-2155. Nadal-Casellas, A.; Proenza, A.M.; Llado, I.; Gianotti, M. Sex-dependent differences in rat hepatic lipid accumulation and insulin sensitivity in response to diet-induced obesity. Biochemistry and Cell Biology 2012, 90, 164-172. Garg, N.; Thakur, S.; McMahan, C.A.; Adamo, M.L. High fat diet induced insulin resistance and glucose intolerance are gender-specific in IGF-1R heterozygous mice. Biochemical and biophysical research communications 2011, 413, 476-480. Hevener, A.; Reichart, D.; Janez, A.; Olefsky, J. Female rats do not exhibit free fatty acid–induced insulin resistance. Diabetes 2002, 51, 1907-1912. Medrikova, D.; Jilkova, Z.; Bardova, K.; Janovska, P.; Rossmeisl, M.; Kopecky, J. Sex differences during the course of diet-induced obesity in mice: adipose tissue expandability and glycemic control. International journal of obesity 2012, 36, 262. Pettersson, U.S.; Waldén, T.B.; Carlsson, P.-O.; Jansson, L.; Phillipson, M. Female mice are protected against high-fat diet induced metabolic syndrome and increase the regulatory T cell population in adipose tissue. PloS one 2012, 7, e46057. Lee, M.-J.; Wu, Y.; Fried, S.K. Adipose tissue heterogeneity: implication of depot differences in adipose tissue for obesity complications. Molecular aspects of medicine 2013, 34, 1-11. Clegg, D.J.; Riedy, C.A.; Smith, K.A.B.; Benoit, S.C.; Woods, S.C. Differential sensitivity to central leptin and insulin in male and female rats. Diabetes 2003, 52, 682-687. Macotela, Y.; Boucher, J.; Tran, T.T.; Kahn, C.R. Sex and depot differences in adipocyte insulin sensitivity and glucose metabolism. Diabetes 2009, 58, 803-812. Jeffery, E.; Wing, A.; Holtrup, B.; Sebo, Z.; Kaplan, J.L.; Saavedra-Peña, R.; Church, C.D.; Colman, L.; Berry, R.; Rodeheffer, M.S. The adipose tissue microenvironment regulates depot-specific adipogenesis in obesity. Cell metabolism 2016, 24, 142-150. Clegg, D.J.; Brown, L.M.; Woods, S.C.; Benoit, S.C. Gonadal hormones determine sensitivity to central leptin and insulin. Diabetes 2006, 55, 978-987. Parks, B.W.; Sallam, T.; Mehrabian, M.; Psychogios, N.; Hui, S.T.; Norheim, F.; Castellani, L.W.; Rau, C.D.; Pan, C.; Phun, J. Genetic architecture of insulin resistance in the mouse. Cell metabolism 2015, 21, 334-347. Wang, S.; Yehya, N.; Schadt, E.E.; Wang, H.; Drake, T.A.; Lusis, A.J. Genetic and genomic analysis of a fat mass trait with complex inheritance reveals marked sex specificity. PLoS genetics 2006, 2, e15. Klein, S.L.; Schiebinger, L.; Stefanick, M.L.; Cahill, L.; Danska, J.; De Vries, G.J.; Kibbe, M.R.; McCarthy, M.M.; Mogil, J.S.; Woodruff, T.K. Opinion: sex inclusion in basic research drives discovery. Proceedings of the National Academy of Sciences 2015, 112, 5257-5258. Becker, J.B.; Prendergast, B.J.; Liang, J.W. Female rats are not more variable than male rats: a meta-analysis of neuroscience studies. Biology of sex differences 2016, 7, 34.

Round 2
Reviewer 1 Report
The response for comment 6 is reasonable. it would be nice if authors could discuss it in the discussion.
Author Response
RE: Combined Soluble Fiber-mediated Intestinal Microbiota Improve Insulin Sensitivity Of Obese Mice (nutrients-679931)
Comment 1: The response for comment 6 is reasonable. it would be nice if authors could discuss it in the discussion.
Response: Thanks for your suggestion. We have properly enriched the discussion on comment 6 (round 1) in the discussion section. Please see lines 398-407 of the revised manuscript.